

# Impact of umbilical cord arterial pH, gestational age, and birth weight on neurodevelopmental outcomes for preterm neonates

Roksana Malak[1], Dorota Sikorska[2], Marta Rosołek[3], Ewa Baum[4], Ewa Mojs[5], Przemysław Daroszewski[6], Monika Matecka[7], Brittany Fechner[2] and Włodzimierz Samborski[2]

[1] Department and Clinic of Rheumatology, Rehabilitation and Internal Diseases, Poznan University of Medical Science, Poznań, Poland, Poznań, Wielkopolskie, Great Poland, Polska

[2] Department and Clinic of Rheumatology, Rehabilitation and Internal Diseases, Poznan University of Medical Science, Poznan, Polska

[3] Department of Physiotherapy, Poznan University of Medical Science, Poznań, Poland, Poznan University of Medical Science, Poznan, Polska

[4] Department of Social Sciences and the Humanities Poznan University of Medical Sciences, Poznan, Polska

[5] Department of Clinical Psychology, Poznan University of Medical Sciences, Poznań, Poland, Poznan University of Medical Science, Poznan, Polska

[6] Department of Organization and Management in Health Care, Poznan University of Medical Sciences, Poznan, Polska

[7] Department of Geriatrics and Gerontology, Poznan University of Medical Sciences, Poznan, Polska

Corresponding author
Roksana Malak, rmalak@ump.edu.pl

## ABSTRACT

**Background**. The aim of this study was to determine the impact of umbilical cord arterial pH, gestational age, and birth weight on neurodevelopmental outcomes for preterm neonates.

**Methods**. We examined 112 neonates. Inclusion criteria were: Saturations greater than 88%, and heart rates between 100–205 beats per minute.

**Measurements**. We assessed several neurodevelopmental factors as part of the Brazelton Neonatal Behavioral Assessment Scale (NBAS), 4th edition, such as asymmetric tonic neck reflex (ATNR), motor maturity, response to sensory stimuli, habituation, and state regulation. Initial assessment parameters such as APGAR score and umbilical cord arterial pH were used to assess neonates.

**Results**. We found a strong correlation between the presence of the sucking reflex and umbilical cord arterial pH ($r = 0.32$; $p = 0.018981$). Umbilical cord arterial pH was also correlated with the presence of asymmetric tonic neck reflex ($r = 0.27$; $p = 0.047124$), cost of attention ($r = 0.31$; $p = 0.025381$) and general motor maturity ($r = 0.34$; $p = 0.011741$).

**Conclusions**. We found that the sucking reflex may be affected in infants with low umbilical cord arterial pH values. Practitioners and parents can use the NBAS to help determine neurodevelopmental factors and outcomes in preterm infants, possibly leading to safer and more effective feeding practices and interventions.

## INTRODUCTION

Feeding problems are one of the most common issues encountered in pediatric care, occurring in approximately one-fourth of healthy children and up to 90% of children at risk for developmental delay (*Barton, Bickell & Fucile, 2018*; *Clawson et al., 2008*). The discharge of infants from hospital neonatal units is often postponed or delayed due to an infant's inability to suck and feed safely (*Lau, 2015*). Although the sucking reflex appears initially in fetal life, not all neonates present a sucking reflex after delivery. Feeding problems in premature infants vary, appearing in 26.8% of low birthweight infants and 40% of premature infants aged between 25–37 weeks gestation (*Harding et al., 2018*). The inability to orally feed can delay discharge from a neonatal care unit (*Poore et al., 2008*).

Preterm infants tend to have problems with the coordination of breathing, sucking, and swallowing, which can lead to respiratory difficulties with feeding (*Gewolb et al., 2001*; *Harding et al., 2018*). According to *Greene, O'Donnell & Walshe (2016)*, an infant's breathing pattern is not only essential for effective sucking but also for the ability to coordinate the muscles of the jaw, lips, tongue, palate, pharynx, upper trunk muscle tone, and normal sensory function. The Neonatal Behavioral Assessment Scale (NBAS) can be used to measure some of the aforementioned abilities. *Erlandsson et al. (2007)* used the NBAS in their study and found that early tactile stimulation by skin-to-skin contact can help develop an infant's prefeeding behavior.

According to *Foster, Psaila & Patterson (2016)* and *Greene, O'Donnell & Walshe (2016)*, nonnutritive sucking allows for a faster transition to nutritive sucking, or full oral feeding, for an infant. Thanks to the diagnosis of neurobehavioral problems including sucking behavior (for instance by, the NBAS), a therapist can plan and apply proper and direct therapeutic interventions such as improving muscle tone, sensory function, and coordination to properly develop sucking and eating. It has been shown that at about the 12th week of gestation a fetus can swallow and at about the 15th week of gestation the fetus can suck (*Reissland et al., 2012*; *Zhang et al., 2017*). Posterior-anterior movements of the tongue and jaw are observed at 18 to 24 weeks gestation and are important for developing the sucking response (*Da Costa, Van den Engel Hoek & Bos, 2008*).

The coordination of sucking and swallowing appears at about 32 to 40 weeks of gestation, while the coordination between sucking, swallowing and breathing appears between 34- and 42-weeks post-conception (*Da Costa, Van den Engel Hoek & Bos, 2008*; *Gewolb et al., 2001*; *Gewolb & Vice, 2006*). However, clinicians and researchers have found that premature infants born before 37 weeks of gestation can be successfully bottle fed, despite neurological immaturity (*Simpson, Schanler & Lau, 2002*). Knowledge surrounding the diagnosis of feeding problems should be improved in order to facilitate proper feeding therapy (*Gewolb et al., 2001*). The oral-motor maturation process correlates with the emergence of motor pathways and the development of the central nervous system. Furthermore, *Kirk, Alder & King (2007)* found that in order to achieve earlier attainment of full oral feeding, they could apply infant behavioral readiness signs and hunger cues to a more general population of premature infants. Feeding readiness is an important part of infant feeding development. Factors such as birth weight and gestational age may relate to feeding readiness and

efficiency and should be investigated and taken into consideration prior to instituting an oral feeding intervention. This is a situation in which a physical therapy intervention might not be recommended due to a neonate's immaturity (*Gewolb et al., 2001*).

Feeding readiness behaviors (FRBs) may have an effect on future feeding efficiency in preterm infants (*White-Traut et al., 2005*). It is crucial for therapists to recognize neurological immaturity, neurodevelopmental delays, and neurobehavioral stability as a foundation for complex neuromotor activity such as oral feeding (*Desai & Lim, 2019*). Strengthening and developing a neonate's neurodevelopment may improve his or her oral feeding ability, but an intervention should be applied after first assessing his or her development and maturity (*Li et al., 2020*). Neonates are classified as preterm infants when early gestational age presents insufficient behavioral abilities and oral feeding difficulties (*Li et al., 2020*). Even if an infant is at about 26 weeks of gestation or older, a child will still improve in his or her neurodevelopment (*Ohnishi et al., 2016*). Other infant characteristics such as congenital heart defects, early surgical intervention, postsurgical complications, difficulties in achieving stability in physiological and behavioral subsystems, poor arousal, muscle tone abnormalities, and poor state regulation may affect a neonate's ability to achieve oral feeding readiness (*Desai & Lim, 2019*).

The sucking reflex is just one of approximately 50 primitive neonatal reflexes (PNRs). Other PNRs include the rooting reflex and jaw jerk reflex which give valuable information regarding the neurological condition of the neonate. However, the sucking reflex has been identified as the most important for feeding. Other neurodevelopmental aspects of neurobehavior such as the asymmetric tonic neck reflex, the Glabellar reflex, the plantar reflex, the Galant reflex, the walking reflex, and the grasping reflex should be also considered during the assessment of a neonate (*Salandy et al., 2019*; *Zafeiriou, 2004*).

However, problems with sucking may be a risk factor for the neonate potentially experiencing adverse neurodevelopmental issues, such as problems with feeding (*Harding et al., 2018*; *Zhang et al., 2017*). Sucking might provide a predictive value for short-term neurodevelopmental outcomes (*McGrath & Braescu, 2004*; *Zhang et al., 2017*). In the first two years of life, despite many factors that may interfere with an infant's feeding behavior, sucking behavior is an early marker of abnormal developmental outcomes (*Wolthuis-Stigter et al., 2015*). It is not only reflexes that are essential for effective feeding. Immature motor skills, hypotonia, an immature neurological system, and sensory development can contribute to underdeveloped sucking skills (*Harding et al., 2018*). Furthermore, the following factors are important when introducing oral interventions with infants: overall development, gestational age, developing swallowing skills before sucking, physiological stability, health status, the development and interpretation of infant oral readiness signs, and early communication (*Dodrill et al., 2008*; *Harding et al., 2018*). The NBAS is able to assess all of the above-mentioned factors. One advantage of the NBAS is that it takes into consideration the behavior of a child, including non-nutritive sucking which should be assessed while preparing an infant to be fed (*Dodrill et al., 2008*; *Kirk, Alder & King, 2007*).

Additionally, proper positioning of the infant plays an important role during feeding time (*Colson, Meek & Hawdon, 2008*). The most recommended position during feeding time is the elevated side-lying position (*Clark et al., 2007*). Further studies may be warranted

regarding proper positioning of the infant, for example using the chin-tuck position. The chin-tuck position and chin tuck against resistance (CTAR) exercise may help in preventing dysphagia because the position enables the proper suprahyoid muscle contraction necessary for the proper working of the tongue (*Park et al., 2020*; *Redstone & West, 2004*; *The Emily Center at Phoenix's Children's Hospital, 2016*; *Yoon, Khoo & Rickard Liow, 2014*). The most important benefit of the chin-tuck position is that it has been widely used to prevent aspiration difficulties in patients with dysphagia. *Ra et al. (2014)* performed a study in order to investigate the effectiveness of feeding and the degree of optimal neck flexion thanks to the chin-tuck position. Aspiration was reduced or eliminated when the chin-tuck position was used with the study's participants which was shown in their Videofluoroscopic Swallow Study (VFSS). The proper position with clear respiratory and digestive tracts (good position of the esophagus and larynx) helped with swallowing in a safe way. The chin-tuck position reduced the distance between the thyroid and the hyoid and the distance between the mandible and the hyoid (*Bond et al., 2020*; *Bülow, Olsson & Ekberg, 2001*; *Ra et al., 2014*; *Versfeld, 2018*; *Welch et al., 1993*). Therefore, the chin-tuck position is important for ensuring that the infant is in a safe position in order to breathe and suck at the same time.

Assessment tools such as the NBAS are recommended to assess an infant's neurodevelopment and potentially find any underlying somatic disorders. Infants who present with somatic disorders may encounter challenges with the sucking reflex due to being less reactive to external stimuli (*Belot et al., 2021*). It should be noted that if the infant being assessed becomes overreactive to stimuli or experiences other medical issues this may interfere with the implementation of the NBAS assessment (*Brazelton & Nugent, 2011*). The aim of this study was to determine the impact of umbilical cord arterial pH, gestational age, and birth weight on neurodevelopmental outcomes for preterm neonates.

## MATERIALS & METHODS

We examined a cohort of 112 infants hospitalized at the Gynecology and Obstetrics Clinical Hospital, all of whom presented with feeding problems. Parents were given informed consent the day of the assessment. All parents provided their written informed consent prior to the assessment of their child. According to the physician's and neonatologist's report, neonates who were included in the study were considered neurodevelopmentally immature and presented problems with feeding. Common comorbidities among study participants included: intraventricular hemorrhage (six patients), bronchopulmonary dysplasia (66 patients), and patients after an intrauterine infection (18 patients). These comorbidities are the reason why they were initially hospitalized. The range of gestational age was wide: between 22 and 40 weeks of gestation, and the gestational age at the assessment ranged between 30 and 41 weeks.

Inclusion criteria included a peripheral oxygen saturation ($SpO_2$) between 88–95%, and a heart rate between 100–205 beats per minute as measured by the Nellcor OxiMax N-600x pulse oximeter. Pulse oximetry was measured on the sole of the foot (*Sink, Hope & Hagadorn, 2011*). Any infants manifesting with clinical instability such as a high temperature, fatigue or deviation of $SpO_2$ or a heart rate outside of the aforementioned

parameters were excluded from the study. The exclusion criteria therefore included desaturation ($SpO_2 < 88\%$), a heart rate <100 or >205 beats per minute, active inflammation, sepsis, bone replacement, tumor, encephalopathy, hypotension and lethal birth defects (*e.g.*, Edwards, Patau etc.) Following the application of inclusion and exclusion criteria, 100 subjects were recruited to the study. Twelve neonates did not meet inclusion crtieria, since their oxygen saturaion was below 88 bpm and their heart rate was heart rate was <100 or >205 beats per minute. Although there were 112 neonates recruited at the beginning of the study, just 100 met inclusion criteria, and 84 of these 100 neonates exhibited insufficient sucking behavior.

We assessed the study's participants using the NBAS just one time in the span of three months, which was not the routine in the neonatal unit. We used the NBAS to check the neonates' maturity and neurobehavior, which including sucking behavior. The average age of gestation was $32 \pm 6$ weeks (minimum 22 weeks postmenstrual age and maximum 40 weeks postmenstrual age). The average gestational age of the infants during the time of the NBAS assessment was $35 \pm 3$ and varied between 30 and 41 weeks, postmenstrual age. The average birth weight was 1,808 g $\pm$ 679 g.

## Measurements

We assessed infant neurobehavior, including sucking, using the Brazelton Neonatal Behavioral Assessment Scale (NBAS), $2^{ND}$ edition as other researchers have done (*Tokunaga et al., 2019*). Our research study did not evaluate the nutritive sucking development for the study's participants due to infants being born premature, not being at the age at which to safely feed orally, and presenting concerns with sucking, swallowing, and breathing coordination (*Gewolb et al., 2001*; *Harding et al., 2018*). We chose to use the NBAS since it takes into consideration the following factors that influence feeding behaviors in infants: habituation, social interaction, motor system, state regulation and organization, autonomic nervous system, reflexes and supplementary items (*Sakalidis & Geddes, 2016*). A researcher who had completed a specialist NBAS assessment course at the Cambridge Brazelton Centre UK performed the NBAS assessment with the study's participants. Each neonate was assessed and then placed on a flat, comfortable surface of an incubator cot. Other parameters considered during the study were umbilical cord arterial pH (collected at birth), birth weight, gestational age at birth, and age at assessment.

## Analytic strategy

Statistical analysis was performed using Statistica Version 13 software (TIBCO Software, Tulsa, USA). The correlation between samples was measured using Spearman's rank correlation, and a *p* value < 0.05 was considered statistically significant. A non-parametric Mann–Whitney *U* test was used to test for group differences in continuous variables.

The study was approved by the Bioethics Committee, consent ref. no. 734/19.

## RESULTS

According to the results from the NBAS assessment as shown in in Table 1, the sucking reflex was present in 84 out of the 100 children assessed. There was a statistically significant

**Table 1  The relationship between sucking and perinatal data.**

| Interview data | r Spearman value | p value |
|---|---|---|
| Gestation age | 0.11 | 0.337654 |
| The age during the assessment | 0.05 | 0.795217 |
| Umbilical cord artery pH | 0.32 | 0.018981 |
| The birth weight | 0.13 | 0.201785 |

**Table 2  The relationship between items of Neonatal Behavioral Assessment Scale and the level of umbilical cord artery.**

| Items of Neonatal Behavior Assessment Scale | p - value | r Spearman |
|---|---|---|
| Asymmetric Tonic Neck Reflex | 0,04 | 0,27 |
| Cost of attention | 0,02 | 0,30 |
| Motor maturity | 0,01 | 0,34 |
| Autonomic system, tremulousness | 0,73 | 0,04 |
| Autonomic system, startless | 0,11 | 0,22 |
| Autonomic system, lability of skin colour | 0,42 | 0,11 |
| State regulation | 0,72 | 0,04 |
| Lability of state | 0,29 | 0,29 |
| Animate auditory | 0,52 | 0,08 |
| Animate visual | 0,50 | 0,09 |
| Inanimate visual | 0,26 | 0,15 |

correlation between the presence of the sucking reflex and the value of umbilical cord arterial pH ($p = 0.018$; $r = 0.32$). However, there was no correlation found between other factors such as gestational age at birth, age during the assessment, birth weight, and presence of the sucking reflex (Table 1). We also observed that umbilical cord arterial pH was correlated with the presence of asymmetric tonic neck reflex (ATNR), cost of attention, and motor maturity (Table 2). Other items from the NBAS such as habituation to sensory stimuli (*i.e.,* Animate auditory, animate visual, and inanimate visual), social interaction, state regulation and organization, as well as autonomic nervous system items and most of the reflexes except for ATNR and other supplementary items did not correlate with umbilical cord arterial pH value and the sucking reflex.

## DISCUSSION

Understanding the sucking reflex in neonates is an important matter, as it forms part of the coordination of sucking, swallowing and breathing essential for breast or bottle feeding (*Sakalidis & Geddes, 2016*). There are many factors that may influence the process of feeding an infant. Factors known to be correlated with a delay in development of the sucking reflex include prematurity or low birth weight (*Kumar et al., 2017*; *Lau et al., 2000*; *Sakalidis et al., 2013*). However, these correlations are not demonstrated in the current study. The reason is that our study is the relatively small size of the study group, therefore

the results need to be confirmed in the future on a larger study group and with multivariate analysis.

Sucking behavior must be coordinated with swallowing and respiration in order to ensure the safe transport of food content from the mouth to the stomach (*Lau, 2015*). Such a complex activity as bottle or breast feeding may be disrupted or unsuccessful following low umbilical cord arterial pH which may lead to hypoxia or acidosis. An umbilical cord arterial pH lower than 7 may indicate poor tissue perfusion and is associated with a higher incidence of intraventricular hemorrhage (IVH) and cystic periventricular leukomalacia (*Levene, Fawer & Lamont, 1982*). An umbilical cord arterial pH lower than 7 can also indicate critical hypoxia, particularly of the brain. Furthermore, this can lead to poor motor outcomes and neurodevelopmental disorders (*Robertson & Perlman, 2006*).

Our analysis indicates a correlation between umbilical cord arterial pH and the presence of the sucking reflex. Umbilical cord arterial pH may be correlated with several neonatal complications (*Mitra et al., 2019*). Primitive reflexes and postural reactions help to assess central nervous system integrity (*Hankins & Speer, 2003*; *Mandich et al., 1994*). Therefore, it is essential to assess the sucking reflex as an element of neurobehavior in infants. Many aspects of infant behavior may influence the sucking reflex, such as gestational age (*Franco et al., 2004*). *Pineda et al. (2019)* showed that stable sucking pressure was related to feeding success only at the 38th week of gestation despite the research group consisting of neonates born before the 32nd week of gestation. In our research group most of the children were younger than 38 weeks of gestation. The average week of gestation was 31,8 (approximately the 32nd week) and the average age during the study was 34 weeks of gestation. This may attribute to the fact that there was no significant difference found between age at birth and age at assessment. Another variable of our research group that we suspected to differ in the presence of the sucking reflex was birth weight. We found that the frequency of sucking behavior increased with gestational age as in the study conducted by *Cunha et al. (2009)*. However, we identified no correlation between sucking behavior and birth weight, as similar to the findings from the study of *Psaila et al. (2014)*.

We suspected that poor reaction to sensory stimuli would be associated with a lack of a sucking reflex. It is well known that preterm neonates tend to present problems with behavioral responses to sensory stimuli (*Rahkonen et al., 2015*). Sucking and swallowing reflexes are complex and involve sensory afferent nerve fibers. The integration of sensory and motor functions is essential to the development of normal sucking and later feeding skills (*Neel et al., 2019*; *Stevenson & Allaire, 1991*). Most of the infants in our study group were born preterm. This is why the assessed response to sensory stimuli as the element of habituation and social interactive items were similar and did not affect the correlation with sucking. However, we know that the development of feeding skills is an extremely complex process influenced by multiple factors, including social factors. Atypical early sensory experiences in the neonatal unit alters others developmental processes such as state regulation (*Litt et al., 2019*; *Neel et al., 2019*). The NBAS enables us to assess an infant's state, which is essential when development feeding programs. The misinterpretation of an infant's state, for instance crying, may lead to problems with feeding including breastfeeding (*Neifert & Bunik, 2013*). However, in the present study most of the neonates were born

preterm and had problems with state regulation. This is why we suspect that the state regulation itself did not affect sucking ability.

Immaturity of the nervous system, including the autonomic nervous system (ANS), characterizes preterm infants (*Malak et al., 2020*). *Franco et al. (2004)* showed that non-nutritive sucking regulated ANS parameters. *Lappi et al. (2007)* demonstrated that sucking behavior imposed physical strain on the baby, with a specific ANS response. However, our study did not show any relationship between sucking behavior and ANS items of the NBAS. The preterm infants in this study very often presented with hyperstimulation of sympathetic ANS. Due to almost all of the infants in our study being born preterm, out study population was homogeneous due to the immaturity of the ANS and poor state regulation.

Asymmetric tonic neck reflex (ATNR) is another component of neurobehavior related to the primitive development of coordination. ATNR should appear at birth or in the first month of life and disappears after six months of life. ATNR is demonstrated with the infant in a supine position. When the head is rotated, extension of the limbs is seen on the side to which the face is turned, and flexion of the limbs is seen on the contralateral side (*Zafeiriou, 2004*). Our study showed a correlation between umbilical cord arterial pH and the appearance of ATNR. Observation of ATNR can also reveal reciprocal activity (*Tokunaga et al., 2019*). This is evidence that adequate neuromuscular coordination is required for ATNR (*Figueras et al., 2011*; *Zafeiriou, 2004*).

Other research has also shown that a relationship exists between automatic postural reactions and motor development (*Mandich et al., 1994*; *Mitra et al., 2019*). Similarly, motor maturity has been shown to be correlated with an umbilical cord arterial pH $\geq 7.00$ (*Figueras et al., 2011*). When an infant is being assessed using the NBAS, is it important to note if movements are smooth and balanced with free arcs (45–90 degrees) or whether movement is jerky with restricted arcs which can indicate immaturity. A higher NBAS score means that the infant presents mature movements (*Tokunaga et al., 2019*). In this study, a higher NBAS score and greater motor maturity was correlated with a higher umbilical cord arterial pH. Other studies have also shown that umbilical cord arterial pH may be a determinant of motor development (*Baschat, 2011*).

Another aspect of NBAS which was correlated with umbilical cord arterial pH was the cost of attention. These are features which may appear in the neonate which should be considered as state overloading. Cost of attention includes observing if the infant is extremely pale or if he or she shows acrocyanosis or mottled skin. Motor exhaustion, flaccidity, hypertonicity and tremors should also be taken into consideration, particularly if an infant's smooth movements turn into jerky and ineffective movements. The correlation between cost of attention and umbilical cord arterial pH is not surprising if we assume that an umbilical cord arterial pH $\leq 7.00$ is a predictor of adverse neonatal outcomes and is an accurate screening test for neonatal hypoxia (*Robertson & Perlman, 2006*).

Infant motor development includes not only reflexes but also perception, planning and motivation. More and more evidence demonstrates that, from birth onwards, babies are subjects who act on the world (*Baschat, 2011*). Delays in motor development, including reflexes, such as sucking, are associated with the presence of neurodevelopmental disorders

(*Diamond, 2000*). *Capilouto et al. (2017)* found that the sucking reflex was a marker for later neurodevelopmental outcomes and is an early marker of overall central nervous system integrity. The lack of an effective sucking reflex was associated with an observed motor developmental delay at 18 months, two years and even three years (*Capilouto et al., 2017*).

### Limitations

One limitation of our study is the relatively small sample size of the study group. Therefore, the results need to be confirmed in the future to a larger study group and with multivariate analysis.

## CONCLUSIONS

Observation of an infant's neurobehavior, including the sucking reflex, is essential to planning and approaching therapeutic feeding interventions, particularly when an infant is discharged home. The NBAS allows the practitioner and parent to observe and assess the abilities and communication of a neonate, and increase his or her ability to engage in appropriate feeding behaviors. This has an added benefit of decreasing the stress of the parent during times when the neonate must feed. Additionally, different aspects should be considered in the assessment of sucking behavior such as ATNR, motor maturity, and cost of attention.

The NBAS enables specialists to appreciate aspects of neurobehavior such as the sucking reflex in the first moments of life and identify any disorders. Clinicians can then communicate findings to the parents and explain aspects that need facilitation. A priority in neonatal care should be allowing parents to spend time with their child in a safe way (*Fonfe, Clements & Mckechnie, 2021*). Moreover, it is important to shorten the length of time a child is separated from his or her family. This is of particular importance during the COVID-19 pandemic when parents cannot be present in the neonatal ward. The NBAS helps parents to identify the neonatal state. Although not all infants who leave neonatal care will go home fully orally fed, the NBAS will support parents in identifying infant states as an important part of oral feeding and oral care development (*Harding et al., 2015*).

## ACKNOWLEDGEMENTS

This study was undertaken with the contribution of nurses and neonatologists from the neonatology clinic who were involved in the preparation of each neonate for neurobehavioral assessment. Special thanks to the nurses and neonatologists at the Gynecology and Obstetrics Hospital in Poznań.

### Funding

The authors received no funding for this work.

## Competing Interests

The authors declare there are no competing interests.

## Author Contributions

- Roksana Malak conceived and designed the experiments, performed the experiments, analyzed the data, prepared figures and/or tables, authored or reviewed drafts of the paper, and approved the final draft.
- Dorota Sikorska and Monika Matecka analyzed the data, prepared figures and/or tables, and approved the final draft.
- Marta Rosołek performed the experiments, analyzed the data, prepared figures and/or tables, and approved the final draft.
- Ewa Baum, Przemysław Daroszewski, Brittany Fechner and Włodzimierz Samborski analyzed the data, authored or reviewed drafts of the paper, and approved the final draft.
- Ewa Mojs performed the experiments, analyzed the data, authored or reviewed drafts of the paper, and approved the final draft.

## Human Ethics

The following information was supplied relating to ethical approvals (i.e., approving body and any reference numbers):

Bioethics Committee at Poznan University of Medical Sciences approved this research (734/19).

## Data Availability

Raw data are available as a Supplementary File.

## Supplemental Information

Supplemental information for this article can be found online at http://dx.doi.org/10.7717/peerj.12043#supplemental-information.

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
