# Peer review of "Impact of umbilical cord arterial pH, gestational age, and birth weight on neurodevelopmental outcomes for preterm neonates"

_PeerJ, doi:10.7717/peerj.12043_

## Round 0.1 · original submission · Major Revisions

Please see reviewers' comments regarding your manuscript. I hope that you will edit your manuscript accordingly and resubmit.

Reviewer 1 ·

Basic reporting

Thank -you for asking me to review this interesting paper, exploring the link between umbilical artery pH and hypoxia by considering infant sucking skills.
Written style: Please have a thorough review of your written English. At times it is challenging to read. I can give a few examples which may help improve the writing, e.g. in the Abstract, line 9; “The initial assessment parameters such as the APGAR score and the umbilical artery pH ….”; in the Introduction, line 18, “Feeding problems are….” May be a better way to start your Introduction section.
References and literature review: I think that you need to revise some of the references used in your text to enable your paper to reflect fully the evidence base available to clinicians specialising in neonatal care.
- You discuss parent absence or reduced access during COVID -19 – please use Fonte et al’s paper to support this statement, as you have no citation. Fonfe, A., Clements, D., & Mckechnie, L. (2021). Parental access to neonatal units: inconsistency during the COVID-19 pandemic. Infant, 17(2).
- Please delete your sentence “The physical and emotional effects of separation should be investigated in further studies” in relation to your COVID statement as it has no bearing on your study, and you do not tie it to any aspect related to your study.
- In relation to the development of sucking and swallowing, I would replace the Lau and the Reissland references you have and use the two Gewolb references here (these are the standard references in most neonatal papers in this area):
Gewolb, I. H., Vice, F. L., Schweitzer‐Kenney, E. L., Taciak, V. L., & Bosma, J. F. (2001). Developmental patterns of rhythmic suck and swallow in preterm infants. Developmental Medicine & Child Neurology, 43(1), 22-27.
Gewolb, I. H., & Vice, F. L. (2006). Maturational changes in the rhythms, patterning, and coordination of respiration and swallow during feeding in preterm and term infants. Developmental Medicine & Child Neurology, 48(7), 589-594.
- Please study your literature review and mention that feeding readiness is an important part of infant feeding development and refer to White -Traut (2005) : White-Traut, R. C., Berbaum, M. L., Lessen, B., McFarlin, B., & Cardenas, L. (2005). Feeding readiness in preterm infants: the relationship between preterm behavioral state and feeding readiness behaviors and efficiency during transition from gavage to oral feeding. MCN: The American Journal of Maternal/Child Nursing, 30(1), 52-59.
- Harding et al (2018) discuss non-nutritive sucking within the oral readiness paradigm – please cite: Harding, C., Cockerill, H., Cane, C., & Law, J. (2018). Using non-nutritive sucking to support feeding development for premature infants: A commentary on approaches and current practice. Journal of pediatric rehabilitation medicine, 11(3), 147-152.
- Not all infants go home fully orally fed, and have a variety of presentations if there are early problems. Please cite: Harding, C., Frank, L., Botting, N., & Hilari, K. (2015). Assessment and management of infant feeding. Infant, 11(3), 85-89.
- You mention positioning. This paper may be useful to cite: Clark, L., Kennedy, G., Pring, T., & Hird, M. (2007). Improving bottle feeding in preterm infants: Investigating the elevated side-lying position. Infant, 3(4), 154-158.

Experimental design

Method: Please state if the parent consent was written / signed. Please additionally state how long parents had written information about the project before agreeing to take part. Please can you present a clear rationale for using the NBAS to assess sucking reflexes? You discuss feeding problems, but you do not discuss anywhere the links between nutritive and non-nutritive sucking development (you can cite Harding et al, (2018) for this).
Data: Your correlations are not clear. In your Abstract, you report only the p value. I think it would be more useful for your readership to use the APA method or reporting correlations. I state this because you Abstract reporting is not clear – and the Spearman values are actually weak, 0.27 for one value, and moderate at .30 and .34 for the other two outcomes. Please re-consider how you will present your data.

Validity of the findings

Applicability: Please discuss how your study can have practical benefit and application for infants, clinicians and families in neonatal care. At present, its practical benefits are difficult to discern.

Additional comments

Please see my comments above. Although this is an interesting area, I feel that the paper needs further revisions and re-writing before it can be useful to the wider neonatal readership.

Reviewer 2 ·

Basic reporting

The aims are reported to be identifying the neurodevelopmental factors associated with sucking. Neurodevelopmental factors, including sucking, were evaluated via the NBAS. The study’s aim was actually to identify how infant characteristics (gestational age at birth, birth weight, gestational age at assessment, and umbilical artery pH) were associated with neurodevelopmental outcomes measured by the NBAS, including sucking. The reporting and title need to better reflect this.

The introduction needs to provide background on all the neurodevelopmental outcomes that were assessed during the study, not just those that turned out to have a significant correlation with pH. The background information on sucking needs to focus more on feeding problems that occur in the neonatal unit – why some infants may have difficulty with sucking and feeding, how this is assessed on the NBAS, how it may present when attempting to feed, and why this is a clinically important problem.

The introduction is brief and needs to provide details of existing literature regarding correlations between infant characteristics and neurodevelopmental outcomes, including feeding.

The introduction includes several sections that are of little relevance to the study, or need to be more clearly linked with what the study aims to find out e.g. these findings are unlikely to help reduce length of stay and get infants home to their parents more quickly, and it is not explained how the study findings would help diagnose feeding problems to facilitate therapy. Additionally, I’m concerned about the suggestion that a chin tuck position may be ‘proper positioning’ – is there any evidence that this would be safe in neonates?

The discussion need to better discuss the findings, including those that were not statistically significant correlations, in relation to what is already known about factors associated with neurodevelopmental outcomes. Do these results agree with what previous studies have identified? Also, how could your findings impact on clinical practice? And what further research is needed to expand and further develop your findings?

Experimental design

As stated above, the stated research aims aren’t consistent with the study as it was carried out. Additionally, there is insufficient detail about the infants included in the study. Other than inclusion/exclusion criteria, how were the infants identified for study? First 100 to meet eligibility criteria? All eligible admissions between certain dates? The authors also need to give more details about the participants e.g. range of gestational ages, gestational age at assessment, proportion of the sample with common comorbidities e.g. intraventricular haemorrhage.

Validity of the findings

Raw data was provided and this shows many participants had missing data e.g. no pH data or incomplete NBAS data. Why was this data not collected and how was this accounted for in the statistical analysis? I would also like to see results for all the components of the NBAS, not just the results that had a statistically significant correlation.

Results need to include confidence intervals as well as, or instead of p-values, as it is not clear if the sample size (16 infants with an absent sucking reflex) is sufficient to reliably determine a correlation with other factors.

Additional comments

These findings are potentially interesting but more work is needed to develop this paper for publication.

---

## Round 0.2 · Minor Revisions

Please see the reviewer's comments suggesting very minor revisions. Looking forward to receiving your revised manuscript soon.

Reviewer 1 ·

Basic reporting

Much improved.

Experimental design

Much improved. However, with Spearman's correlation, please make the R you have small, as r, and please make sure p is in lower case and italicised.

Validity of the findings

Much improved.

Additional comments

This paper has been much improved by the changes.

---

## Round 0.3 · accepted · Accept

Thank you for your speedy reply and congratulations on acceptance of your manuscript!